# An Evolving HIV Epidemic in the Middle East and North Africa (MENA) Region: A Scoping Review

**DOI:** 10.3390/ijerph20053844

**Published:** 2023-02-21

**Authors:** Arvin Karbasi, Judy Fordjuoh, Mentalla Abbas, Chukwuemeka Iloegbu, John Patena, Deborah Adenikinju, Dorice Vieira, Joyce Gyamfi, Emmanuel Peprah

**Affiliations:** 1Global Health Program, Department of Social and Behavioral Sciences, ISEE Lab, NYU School of Global Public Health, 708 Broadway, 4th FL, New York, NY 10003, USA; 2NYU Health Sciences Library, NYU Grossman School of Medicine, 577 First Avenue, New York, NY 10016, USA

**Keywords:** HIV, key populations, people who inject drugs (PWID), Middle East and North Africa (MENA), under-reporting, HIV data

## Abstract

Human immunodeficiency virus (HIV) in the Middle East and North Africa (MENA) region is severely understudied despite the region’s increase in new HIV infections since 2010. A key population that is particularly affected, due to the lack of adequate knowledge and proper interventional implementation, includes people who inject drugs (PWID). Furthermore, the paucity of HIV data (prevalence and trends) worsens an already critical situation in this region. A scoping review was conducted to address the scarcity of information and to synthesize the available data on HIV prevalence rates within the key population of PWID throughout the MENA region. Information was sourced from major public health databases and world health reports. Of the 1864 articles screened, 40 studies discussed the various factors contributing to the under-reporting of HIV data in the MENA region among PWID. High and overlapping risk behaviors were cited as the most prevalent reason why HIV trends were incomprehensible and hard to characterize among PWID, followed by lack of service utilization, lack of intervention-based programs, cultural norms, lack of advanced HIV surveillance systems, and protracted humanitarian emergencies. Overall, the lack of reported information limits any adequate response to the growing and unknown HIV trends throughout the region.

## 1. Introduction

The Middle East and North Africa (MENA) region has been a blind spot throughout existing HIV research due to the lack of HIV data on populations living in this region [1,2,3]. The dearth of data on HIV serves as a critical barrier to achieving the goals set by the joint United Nations (UN) program on HIV/AIDS’ fast track agenda, which includes a “90–90–90” agenda to: (i) diagnose 90% of people living with HIV (PLWH); (ii) provide anti-retroviral therapy (ART) for 90% of those diagnosed; and (iii) achieve viral suppression for 90% of the treated [4]. Although the MENA region has made advancements in HIV research, significant gaps persist, including a lack of HIV data, which impedes comprehensive therapy for many populations within the treatment cascade [5]. Therefore, to end the HIV/AIDS epidemic by 2030, it is essential for key populations at higher risk of infection to have access to comprehensive prevention and treatment services [6]. 

According to UNAIDS data, the unique aspects of the HIV epidemic in the MENA region are ambiguous compared to other regions of the world [7]. HIV infections are still growing, but there is relatively slow growth of data despite a welcome wave of humanitarian efforts and greater integration of HIV and related health services within the region [8]. The dynamic balance of several competing and offsetting factors—including new infections of key populations, migratory patterns of refugees, and discontinuation of HIV services in conflict-ridden regions—meaningfully contribute to the lack of HIV infection control [9,10]. Currently, according to the UNAIDS 2021 report, only 38% of PLWH have access to treatment services [8]. Yet, in contrast to the slow regional progress demonstrated by other countries within the region, Iran and Morocco are outliers and have reported that the annual new infections within their countries have declined by more than 10% since 2010 [8,11,12,13]. This is due in part to the countries’ more advanced HIV surveillance systems and the sustained focus addressing HIV/AIDS within their respective countries. Access to HIV testing, treatment, and care in the whole MENA region remains well below the global average, with only 52% PLWH aware of their status, 38% accessing ART, and less than 40% virally suppressed [8,11,13].

The MENA region is home to several major drug trafficking routes and has been a major transit point of illicit drugs. As a result, this has had an indirect causal effect on the HIV/AIDS epidemic within the region by increasing access and usage of illicit drugs [14]. For example, people who inject drugs (PWID) are most at risk of HIV infection, with more than 20% of annual new infections being attributed to PWID and their sexual partners [11]. Additionally, patterns of illicit drug use and HIV infections among PWID have fluctuated from 2014 to 2019, with an increasing number of new infections. The PWID population is estimated to be between 626,000 to 887,000 individuals, with the HIV prevalence concentrated on these key populations [8,15,16,17,18]. However, the extent to which this population has been affected has only been speculated and modeled, and the prevalence of HIV is assumed to be higher. 

Evidence exists of concentrated epidemics among PWID in Iran and Morocco, and these concentration trends are regionally specific [16,17]. In Iran, HIV prevalence is higher in eastern cities such as Kerman due to the proximity of these cities bordering Afghanistan and being a main drug trafficking route [19]. With the successes that Iran and Morocco have demonstrated in targeting and curbing HIV transmission among PWID, the two countries serve as models that other countries can emulate for HIV intervention and policy implementation. Despite promising observations from Iran and Morocco in controlling the HIV epidemic among PWID, the region lacks sufficient evidence to determine the actual size and HIV prevalence in this population. This reflects the incompleteness of HIV surveillance systems and the fragility of the infrastructure developed to monitor HIV trends. Given that PWID account for approximately 25% to 33% of new HIV infections within the region, this deficiency must be addressed so that more effective and focused programs can be implemented [8,11,20]. 

Assessing the impact of the dearth of HIV data on PWID in MENA is both timely and critical [21,22]. While past reviews have examined the prevalence and incidence of the HIV epidemic among PWID within the MENA region, there exists no comprehensive scoping review focused exclusively on understanding both the availability of HIV data as well as contributing factors to the under-reporting of HIV within the region. The purpose of this scoping review is to establish the epidemiological risk factors and underlying factors contributing to this lack of quality HIV data. This scoping review aims to shed light on existing gaps in the literature and to provide evidence to inform future development of policies and interventions to target people living with HIV who are also injection drug users in MENA. 

## 2. Materials and Methods

A scoping review was conducted to synthesize the available data on HIV prevalence rates within the PWID key population throughout the MENA region. The aim was to address the under-reporting gap of HIV/AIDS within this population. This review was guided by the “Arksey and O’Malley” framework and the Preferred Reporting Items for Systematic Review and Meta-Analysis (PRISMA) checklist to ensure a comprehensive appraisal of eligible articles [23,24].

### 2.1. Inclusion/Exclusion Criteria

Studies were included if they met the following criteria: (i) conducted in the MENA region; (ii) included PWID; (iii) addressed HIV prevalence and treatment; (iv) addressed under-reporting of HIV data, through either quantitative or qualitative studies; (v) published in English or had translated versions available for non-English articles; and, (vi) the study took place between 2007 and 2022. No limitations were placed on the study design or type of articles assessed. Studies were excluded if they: (i) excluded PWID; (ii) did not address factors contributing to under-reporting; (iii) focused on HCV, HBV, tuberculosis, and other communicable diseases; (iv) were not conducted in the MENA region; and (v) included no measure of HIV infection rate in PWID.

### 2.2. Literature Search Methods

The literature search was conducted in the following databases: PubMed, MEDLINE, Embase, APA PsycInfo, Cochrane Library, Global Health (Ovid), Web of Science (all databases); and the Cumulative Index to Allied Health Literature (CINAHL). Gray literature from the following databases were used: UNAIDS Report, World Drug Report, Harm-Reduction Advocacy Brief for the MENA region, abstracts from the International Aids Society (IAS); and posters from international conferences. All citations were managed using EndNote 20, a bibliographic management program.

A comprehensive search strategy was developed to identify studies that met the predefined inclusion criteria. Multiple MENA countries were included, as defined by the UNAIDS (i.e., Algeria, Bahrain, Egypt, Iran, Iraq, Jordan, Kuwait, Lebanon, Libya, Morocco, Palestine, Oman, Qatar, Saudi Arabia, Sudan, Syria, Tunisia, United Arab Emirates (UAE), and Yemen). The literature search was conducted on 29 April 2022. The full search strategy is provided in Appendix A.

### 2.3. Assessment

All citations were downloaded to Covidence for screening. Retrieved articles were reviewed and assessed separately by two pairs of reviewers to reduce bias. Each pair of reviewers screened the same articles independently to determine if the inclusion criteria were met by assessing the title and abstract. Title abstract screening, assessment, and extraction were conducted by A.K., M.A., J.F. Articles were randomly assigned where two would screen and third would resolve disagreements. After confirming that the preselected articles met the criteria, a Google Form (see Appendix B) was utilized to extract the data. Factors contributing to underreporting were collected and tabulated in an excel form. Inconsistencies in data were subjected to the judgement of a third independent reviewer. Final articles were ultimately chosen by consensus. Data were collected on the type of study design, the aim of the study, geographic location, sample size/number of participants, length of the study, gender reported, age range of the population, racial/ethnic group, income range of the country as indicated by World Bank standards [25], high and overlapping risk behaviors, gaps in healthcare access and HIV testing, protracted humanitarian emergencies, stigma and discrimination, lack of advanced surveillance systems, and the lack of intervention-based programs focused on PWID, population estimates of PWID, HIV prevalence in PWID, suggestions for effective interventions/programs, and estimated number of new infections resulting from PWID. 

### 2.4. Extraction

Data were extracted as follows: (i) a descriptive table of the included studies and location where the study took place; (ii) a table narrating the state of the epidemic (e.g., concentrated, emerging, low-level, etc.), HIV-prevalence in PWID and amount of evidence supporting the finding; and (iii) reporting of descriptive analysis, such as population and HIV prevalence estimates from 2007 to 2022. Studies were reviewed for reporting on under-reporting directly or indirectly regarding HIV among PWID within the MENA region. 

## 3. Results

An initial search retrieved 2955 articles, with 1864 articles remaining after removing duplicates. Screening of title and abstract further excluded 1381 articles, with an additional 331 duplicates removed. In total, 152 full-text articles were assessed, of which 38 studies and 2 reports were extracted and included in the results. The screening, elimination process, and reason for excluding articles are outlined in the PRISMA chart (see Figure 1). 

In total, 40 studies were conducted across 20 MENA countries, with Iran as the leading location appearing in 34 studies. Of the 40 studies, 28 cited high and overlapping risk behaviors as a contributing factor to the under-reporting of HIV data in the MENA region, followed by a lack of service utilization in 28 studies, a lack of intervention-based programs targeting PWID reported in 26 studies, cultural norms in 24 studies, a lack of advanced HIV surveillance systems in 17 studies, and protracted humanitarian emergencies reported in 2 studies (see Table 1).

### 3.1. High and Overlapping Risk Behaviors 

High and overlapping risk behaviors among PWID within the MENA region play an instrumental role in the spread of HIV and are major contributors to the discrepancy in the region’s HIV data. High-risk practices, especially within PWID, expose them to infectious diseases (e.g., HIV, HCV, HBV, TB, etc.) through contaminated injection equipment and unprotected sex [47]. Specifically for HIV, the presence of multiple exposure routes poses an issue for proper and adequate targeting of prevention and treatment. There is significant overlap between different key populations, where some PWID are sex workers, buy/trade drugs for sex, or are men who have sex with men (MSM) and who engage in drug-injecting practices and unprotected sex [29,42,47]. 

The reporting of high and overlapping risk behaviors significantly contributes to the discrepancy of available data within the region and remains the most cited issue in the studies. A significant portion of PWID exhibited high-risk behaviors including, but not limited to, having multiple sexual partners with inconsistent condom use, the sharing of injecting equipment, and needle and syringe sharing between partners. A study assessing male drug injection behaviors revealed that study participants engaged in drug injection at a rate of approximately one to two injections per day [2]. Moreover, the study found that approximately 50% of PWID within the MENA region have at one point shared injecting equipment, with the use of clean injecting equipment being limited [2]. Other high-risk behaviors observed among PWID include but are not limited to: (i) drawing blood back into the syringe; (ii) selling sex for money or favors; and (iii) having multiple sexual partners^2^. Various studies investigating risky behaviors in female PWID revealed that study participants exhibit a riskier injection profile than their male counterparts, which could be partially attributed to the population’s greater experience of stigmatization and difficulties in accessing harm-reduction services [28,46].

In another study, which conducted a national bio-behavioral survey in Cairo, Egypt, between 2013 and 2017, investigators found that certain sociodemographic factors played an instrumental role in the frequency of certain high-risk behaviors being exhibited more than others [27]. The survey revealed that more than three-fourths (i.e., 86%) of male PWID reported reusing needles or syringes, and just under three-fourths (i.e., 74%) reported ever sharing needles or syringes, thereby indicating that risky injecting practices among PWID in Cairo remains a major concern [27]. Among studies investigating adolescents and young adults (i.e., those 25 years and younger) in Iran, there was a higher incidence of HIV in younger populations, which could be explained by the higher frequency of engaging in high-risk sexual acts, such as multiple sexual relationships, unprotected sex, alcohol use, and/or unsafe drug injections with multiple injecting partners [44,48]. In another cross-sectional study design in Libya, 85% of PWID reported sharing needles previously, with up to 29% doing so within the past month—thereby revealing the prominence of needle sharing within the region [37]. Additionally, of the individuals participating in the Libya study, only two-thirds reported having used a condom during sexual practices [6]. 

In addition to the sharing of injection equipment being a major mode of HIV transmission, the percentage of transmission of HIV due to unsafe sex has seen a considerable increase from 1984–2015 [5,49]. In Oman, from 1984 to 2015, nearly 66.8% of HIV transmission occurred through unsafe sex, whereas from 2015 to 2018, the number has risen to higher than 88% [5,49]. Another study observed that certain demographic characteristics contributed to increased odds of unsafe sexual and injection drug use practices, such as being married or a younger injection drug users (IDU) [28].

### 3.2. Stigma and Discrimination 

Of the twenty (20) MENA countries studied, fourteen have strict punitive laws criminalizing the possession or use of small amounts of drugs [8,11,20]. The repressive and inhumane approaches to drugs—such as corporal punishment and the death penalty—only dissuade PWID from seeking the proper preventative and necessary treatment options at health care centers. Additionally, there is a reluctance to acknowledge HIV in PWID; and with strict legislation in place, the marginalization and stigmatization of PWID is reinforced, and high and overlapping risk behaviors to avoid corporal punishment and negative stereotypes is perpetuated [8,11,26].

Within the MENA region particularly, both male and female PWID experience varying levels of stigma. Additionally, female PWID are highly stigmatized and are considered one of the hardest-to-reach populations, which explains the lack of data on this specific key population [16,26,31,46,50]. Moreover, PWID, as a population, seem to be reluctant to seek or access HIV services, such as needle/syringe program (NSP) and opioid agonist treatment (OAT) [26,27,31,50]. In a study investigating stigmatizing attitudes in hospitals within Egypt, it was found that there a significant number of stigmatizing attitudes by health professionals, which may result in depriving patients of their health rights and can dissuade people from disclosing their HIV status [32].

While Iran has been successful in curbing the HIV epidemic within its PWID key population, the country still faces barriers as it pertains to HIV care. The socio-cultural stigma within several communities across the country creates barriers to accessing essential health and harm-reduction services [20,51]. Additionally, HIV was, and is still to a certain extent, assumed to be less prevalent in the MENA region due to the dominant practice of Islamic rituals and related socio-cultural settings [33]. In another study investigating the individual, community, and structural-level stigmas within MENA, it was observed that among youth, there was a decrease in HIV testing leading to a lack of HIV diagnosis within the region [15]. Various studies have cited that the sensitivity toward HIV in PWID restricts access to HIV-related data and poses additional challenges to surveillance and further research efforts being conducted within this region [20]. 

Most studies have cited that the severe punitive approach to curb the HIV epidemic has not only exacerbated the HIV crisis but has also led to a genesis of widespread stigma toward, and a lack of support for, various treatment programs, compounded by the internalized fear and stigma associated with having HIV and being an injection drug user [26]. This is best exemplified in Egypt, where PWID are highly stigmatized. This partially explains the drop in HIV prevalence (approx. 4.4%) observed relative to the HIV prevalence (approx. 2.5) seen in a 2016 biobehavioral survey among PWID [27,39,41]. Furthermore, the legal and personal repercussions, coupled with various factors including, but not limited to, negative perceptions from healthcare professionals and high medical costs, have contributed and will continue to influence the denial of infection and identification of disease. 

### 3.3. Lack of Intervention-Based Programs Targeting PWID and Service Utilization

Fourteen countries have mentioned harm reduction and PWID in their national policy documents. Of those fourteen (14) countries, five (5) (Egypt, Iran, Lebanon, Morocco, and Oman) adopted harm reduction policies in their national AIDS strategic plan, and seven (7) (Algeria, Bahrain, Jordan, Libya, Oman, Syria, and Tunisia) highlight PWID as the key population to target [8,11]. According to the UNAIDS report, the number of countries providing NSP has increased; however, the coverage of the harm-reduction services remains insufficient [43].

Iran has served as a model for implementing progressive HIV care interventions targeting PWID. In Iran, the initiation of triangular clinics integrating services for treatment and prevention of injection drug practices has contributed to the decrease in HIV incidence within the PWID key population. This contribution is in addition to other interventions implemented within Iran, such as: (i) drop-in centers; (ii) integration of substance-use treatment; and (iii) HIV prevention in rural primary health care systems [52]. Yet, despite Iran’s best efforts to integrate harm-reduction services into their healthcare system, HIV testing uptake of PWID remains low [45]. 

As for other MENA countries, there remains an urgent need to emulate Iran’s controlling and monitoring of the current and unknown epidemic within their own respective key populations. Iran remains a leader in the implementation of harm reduction with an estimated net coverage of approximately 55–77% among PWID (as of 2014), with Morocco coming in second as one of the only few countries with an operational OAT and NSP program [10]. Given the similarities between various MENA countries as it pertains to socio-cultural practices and beliefs, the countries stand to benefit from Iran and Morocco’s success in the implementation of culturally tailored, harm-reduction services into their respective healthcare systems. Moreover, it is reasonable to assume that a stable political regime in both Iran and Morocco played a role in ensuring the proper development of an acceptable harm reduction program—a necessary condition precedent that is absent in most other countries within the MENA region. Moreover, with the age of first injection progressively getting younger, harm-reduction within MENA should begin targeting and tailoring toward a younger demographic and be linked to other sectors of health care to mitigate, and hopefully, eliminate, the growing epidemic [10].

Several studies have noted the dearth of data pertaining to treatment adherence and linkage within the MENA region. According to the MENA Harm-Reduction advocation briefing report, only 38% of PLWH are accessing treatment and prevention services. This coverage is not only lacking behind other regions throughout the world, but is also considerably short of reaching the UNAIDS 2025 AIDS targets of 95% of PLWH on ART. In a study focused on Sudan’s efforts to improve linkage to and retention in care, there was an implementation of a search and rescue strategy to actively identify hard-to-reach key populations to reduce attrition [5,53]. However, this proved to not be beneficial, as the lack of data within this region ultimately prevented the adequate development of a program to reach the target key population. Clearly, there is an urgent need to adopt strategies to enhance linkage and adherence to care through targeting approaches of certain key populations [40]. More importantly, according to the UNAIDS regional report, PWID have accounted for nearly 25% of new HIV infections in the area, thereby making them a key population of interest [11]. As alluded to previously, improving outreach to PWID entails revisiting current punitive legislation and developing socio-culturally tailored programs aimed at alleviating sensitivities related to this key population [20]. 

Studies have also noted the limited access of needles, syringes, and condoms as major gaps in HIV prevention across all different subgroups of PWID. The inaccessibility of these services poses serious threats to harm reduction programs tailored toward PWID, and it restricts the potential existing services in curbing the HIV epidemic within the MENA region. Despite the availability of free-of-charge programs—such as NSPs, OAT, and condom distribution services in Iran—PWID continue to engage in high-risk practices due to the poor quality of services of pre-existing programs [30].

PWID have low perceived risk of HIV, high stigma around drug injection and use, and low access to harm reduction education and prevention services. Consequently, the effective implementation of intervention programs may prove to be beneficial in enhancing the transmission of knowledge and service utilization within this key population. 

### 3.4. Protracted Humanitarian Emergencies 

The continuing global economic crisis and lack of resources within certain areas, compounded by continuing conflicts within the region, prevent MENA countries from prioritizing internal issues above external factors [33]. The addition of forced displacement of civilians combined with the socio-political issues occurring within and between countries has also created one of the worst humanitarian crises of time [33]. In the last 10 years, several countries have been directly and indirectly involved in conflicts and civil wars [33,34]. Consider Syria, where approximately 8.7 million people are estimated to be displaced internally, with 4.8 million having left Syria as of 2015 [26,33,54,55]. In Iraq, as many as 3.9 million Iraqis were internally displaced by mid-2015 [33,55]. Therefore, estimating the size of the population or defining the population, which are key indicators in calculating HIV rates, remains very challenging and results in conflicting data within the region [56]. Furthermore, forced displacement may contribute to the vulnerability of refugees to HIV, as evidenced by the positive correlation in conflict-ridden areas such as Syria and Iraq [26]. Countries that have faced humanitarian crises have endured institutional degradation and a loss of human capital, which results in severe disruptions in HIV-focused healthcare services as well as weak HIV surveillance programs [20,22].

The compounding of various factors on the already fragile healthcare systems disrupts HIV programs and services as well as accessibility to the services. Consider, once again, Syria, where the National AIDS program estimates up to a 99% reduction in HIV surveillance efforts among key populations—including PWID and female sex workers—following the onset of conflict [20]. Furthermore, the services being administered are further impeded by the lack of voluntary counseling and testing capacity, which threatens to yield serious implications for public health response planning. In Lebanon and Yemen, services and surveillance efforts have been non-operational due to the ongoing humanitarian crises. In Yemen’s case, clinicians have been prevented from administering care to PLWH [20,34].

### 3.5. Lack of Advanced Surveillance Systems 

With the paucity of HIV data, including the lack of integrated biological and behavioral surveillance surveys (IBBSS) within the health infrastructure, PWID is particularly evident in eight (8) countries: Algeria, Qatar, United Arab Emirates, Iraq, Djibouti, Somalia, Sudan, and Yemen [53]. Furthermore, to develop and implement an adequate response to the growing epidemic, knowledge of the status of the HIV status within the region is needed. 

HIV prevalence in PWID was estimated in eleven (11) MENA countries (see Table 2) and, of these, Iran, which also recorded one of the highest levels of HIV prevalence in PWID, demonstrated trend data of a decrease in the country’s HIV prevalence [11]. This noticeable shift within Iran further strengthens the previous assertion that adequate knowledge of the HIV trends within a country is essential to curbing the growing epidemic [12]. Independent of Iran, more than one-third of the countries still do not have any IBBSS data [11]. This inaction is to a large extent attributed to the lack of political will, denial, social stigma, and discriminatory policies—including the criminalization of drug use [57]. Additionally, the inability of countries to allocate funds to HIV programs to further improve HIV surveillance systems within the region impede an adequate HIV response to improve HIV outcomes in PWID [53].

However, in some parts of the region, HIV surveillance activities have been sustained for a period to keep track of the HIV trends in key populations. Surveillance data have shown a decline in HIV incidence and prevalence within PWID in Iran, which could be explained by the extensive harm-reduction programs that have been implemented in the country [44,57,58]. However, in other countries, the status of HIV in PWID is either unknown or has no sign of slowing down. 

## 4. Discussion

To the best of our knowledge, this scoping review is the first to examine the predominant lack of research focused on understanding the factors contributing to the lack of quality HIV data on PWID in the MENA region. Moreover, this review summarizes the current evidence and provides an updated understanding of the trends within the MENA region regarding PWID and steps forward in addressing these key issues contributing to the growing and unknown HIV epidemic throughout the MENA region. 

We found that most studies were conducted in Iran, with a few in other countries within the region (see Figure 2). Where HIV prevalence data were available for all key populations, the HIV epidemic was largely concentrated in PWID (see Table 3). This finding is consistent with previous studies included in this review among PWID within the MENA region. Additionally, HIV prevalence ranged from 0% to 14% in PWID and was available for eleven of the twenty countries investigated (see Table 2). However, due to some of the data being outdated or first estimated years ago, it is possible that the prevalence rate may be much higher than noted or for the epidemic to spread to other key populations within the region. These findings provide insight into the growing epidemic within each country and the key population that is most at risk of acquiring and spreading HIV to other key populations. Another factor contributing to the growing epidemic and lack of up-to-date HIV information for this population is the stigmatization of this key population. 

PWID are considered a highly stigmatized and hard-to-reach key population within the region. Across all MENA countries included in this scoping review, it is evident that overlapping risk behaviors—such as low condom use and sharing of injection equipment—is very common among this group. In addition, endemic to certain areas within the region seems to be the sparsely documented HIV trends occurring within the key population. 

In addition to the stigma associated with this key population, the humanitarian emergencies occurring within the region contribute to an already fragile health infrastructure. These events impede access to HIV treatment by impacting HIV incidence, further spreading the possibility of transmission between different countries or groups of individuals, making it difficult to gather accurate HIV data on population sizes and prevalence. Most at risk are mobile populations that endure poorer living conditions and limited access to HIV treatment services. The migration rate of individuals within MENA is unusually higher than any other region in the world due to recent conflicts that expand the displacement of refugee populations, thereby contributing to the lack of accurate HIV data within the region [20,59]. Finally, the socio-political issues within the region contribute to the lack of resources in the region by diminishing access to services. Harm-reduction services such as NSP, OAT, and condom distribution may be disrupted and discontinued entirely during humanitarian crises.

With various socio-political conflicts within the region, HIV is de-prioritized because of competition from other urgent priorities [60]. Furthermore, the weak operational capacity of organizations with PWID have limited procedures and protocols regarding the provision of treatment services and monitoring of the HIV trends in key populations. 

Intensified surveillance and massively expanded HIV testing will be key to controlling the epidemic. The failure to capture key data on risk behaviors, improve treatment adherence among PWID, and the introduction of drug resistance from unknown populations is a missed opportunity to further understand epidemic. Such data can inform policies to lower HIV incidence and mortality by helping target resources to where the largest impact could be expected [38]. 

Building on successful local programming is essential to improving the HIV infrastructure within the region. The funding and scalability for HIV programs is essential to reaching hard-to-reach key populations. Within the region, especially in conflict-ridden areas, there is high staff turnover and major disruptions in HIV testing and volunteering efforts for HIV programs [60]. These realities make it difficult for key populations to access treatment areas, which contribute to the lack of testing and impede the MENA region from reaching the UNAIDS goal of “90–90–90”. The improvement of the fragile infrastructure within the region through partnerships with non-governmental organizations (NGOs) to provide comprehensive care is another factor that can increase testing and impede the growing HIV epidemic [11,41,60]. The establishment of such a partnership may influence the perception of the key populations requiring the service in a positive fashion and enable key populations to access HIV services [9,60].

Finally, the development of a central mechanism to report on HIV trends occurring within the region is essential to overcoming the discrepancies in HIV data. The MENA region suffers from misinformation and accurate data to inform future policy decisions and interventional implementations [36,60]. This scoping review highlights the challenges within the MENA region as it pertains to the documentation of HIV trends occurring within PWID to encourage further public health research and policy work investigating regions where data are significantly inaccurate or missing to encourage and facilitate further action. The availability, access, and increasing awareness toward the pressing issue in the MENA region are crucial in addressing the growing epidemic affecting this key population. However, with MENA, we note two exemplary countries with strong infrastructure for surveillance and reporting for PLWH, including Iran and Morocco. Iran seems to be especially proficient to combine its data with relatively high HIV testing trends among its population [8,10,11]. For instance, in 2017, Iran scaled up harm-reduction services considerably and distributed 8,578,845 needles/syringes, which is an average of about 43 needles/syringes for each IDU. Additionally, the number of needle exchange program sites also saw a considerable increase with an addition of 548 new sites between 2008 and 2017 [10]. Iran’s ability to expand harm-reduction services was imperative to slowing down and curbing the HIV epidemic within their PWID population and may be a key factor as to why they have decreased HIV prevalence within this key population.

## 5. Conclusions

Based on this research, there is a predominant lack of research focused on understanding the lack of quality HIV data on PWID in the MENA region. Given the paucity of data on the key populations, the information gathered allows for the identification of key factors that must be addressed to inform the development of future regional policies and programs.

The data synthesized from this review were used to provide an updated classification of the state of the HIV epidemic in all the countries within MENA for which data were available. These findings suggest that PWID play an instrumental role in the spread of HIV infections due to their high-risk practices and behaviors. The window of opportunity to control the epidemic should not be ignored, and it is essential that HIV care and prevention among PWID be made a priority in this region. Furthermore, there is a need to further understand the current state of the epidemic of all countries within the region to appropriately structure existing HIV programs and national policies to curb the budding and growing epidemics occurring within this key population. The improvement of HIV programs within MENA is not only essential to the PWID population but will also confront the growing HIV problem in other key populations, where the HIV epidemic is not yet known.

## Figures and Tables

**Figure 1 ijerph-20-03844-f001:**
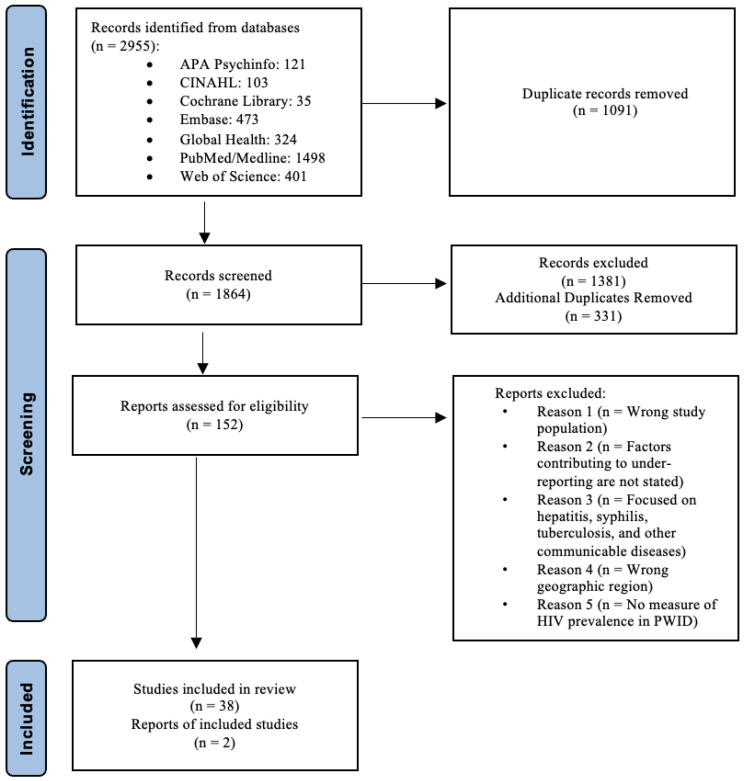
Flow diagram of study selection.

**Figure 2 ijerph-20-03844-f002:**
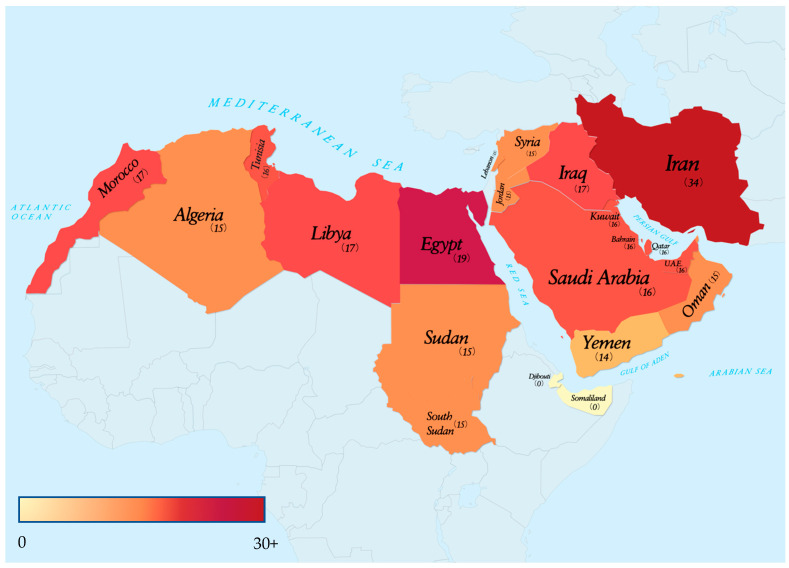
Heat map representing the number of research articles found for each country. The gradient bar scaled 0–30+, indicates the number of research studies found to have been conducted in the country.

**Table 1 ijerph-20-03844-t001:** List of included studies attained from the assessment and screening process.

Author (Year)	Country	Study Objective	Study Type	Income Level	Factor Contributing to Under-Reporting (UR)
Aaraj et al. (2016) [26]	All Countries	The objective of the study was to highlight civil society as principal partners of government in scaling up the response to HIV and in implementing national HIV policies.	Systematic Review and other Literature Reviews	NA *	High and overlapping risk behaviors, Lack of service utilization (e.g., gaps in health care access, HIV testing, etc.), Cultural Norms (Stigma and Discrimination)
Abu-Raddad et al. (2010) [2]	All Countries	The objective of this review and synthesis was to address the dearth of strategic interpretable data on HIV in MENA by delineating a data-driven overview of HIV epidemiology in this region.	Systematic Review and other Literature Reviews	NA *	High and overlapping risk behaviors, Lack of service utilization (e.g., gaps in health care access, HIV testing, etc.), Cultural Norms (Stigma and Discrimination), Lack of intervention-based programs targeting PWID (Harm-reduction)
Anwar et al. (2021) [27]	Egypt	The objective of this study was to assess whether sociodemographic factors were associated with injecting and sexual behaviors in Egypt among people who inject drugs.	Non-Randomized Control	Lower-Middle income	High and overlapping risk behaviors, Lack of service utilization (e.g., gaps in health care access, HIV testing, etc.), Cultural Norms (Stigma and Discrimination)
Esmaeli et al. (2019) [28]	Iran	The objective of this study was to address the gap of the frequency of both dual and single risky behaviors among PWID.	Qualitative study	Lower-middle income	High and overlapping risk behaviors, Lack of service utilization (e.g., gaps in health care access, HIV testing, etc.), Cultural Norms (Stigma and Discrimination), Lack of advanced HIV surveillance systems (no infrastructure to properly monitor HIV disease progression), Lack of intervention-based programs targeting PWID (Harm-reduction)
Etemad et al. (2020) [29]	Iran	The aim of this study was to determine the prevalence of HIV infection and associated risk behaviors among male PWIDs in Kermanshah city in 2017.	Non-Randomized control	Lower-Middle income	High and overlapping risk behaviors
Gangi et al. (2020) [30]	Iran	The aim of this study was to estimate the HIV prevention cascades among PWID in Iran.	Non-Randomized Control	Lower-middle income	High and overlapping risk behaviors, Lack of service utilization (e.g., gaps in health care access, HIV testing, etc.), Cultural Norms (Stigma and Discrimination), Lack of intervention-based programs targeting PWID (Harm-reduction)
Hashemipour et al. (2013) [19]	Iran	The aim of this study was to determine HIV prevalence among individuals with a background of intravenous drug use via community announcement in Isfahan, Iran.	Non-Randomized Control	Lower-Middle income	High and overlapping risk behaviors
Heijnen et al. (2016) [14]	All Countries	The objective of this review was to synthesize information from available HIV and HCV data in prisons in MENA and highlighted opportunities for action.	Systematic Review and other Literature reviews	NA *	High and overlapping risk behaviors, Lack of service utilization (e.g., gaps in health care access, HIV testing, etc.), Lack of intervention-based programs targeting PWID (Harm-reduction)
Jamshidimanesh et al. (2017) [31]	Iran	The objective of this study was to understand the perceptions of Iranian drug-dependent females with high-risk behaviors for HIV/AIDS.	Qualitative study	Lower-Middle income	High and overlapping risk behaviors, Lack of intervention-based programs targeting PWID (Harm-reduction)
Kabbash et al. (2018) [32]	Egypt	The aim of this study was to assess attitudes of health care providers toward PLWH at Tanta University Hospitals in Egypt.	Non-Randomized Control	Lower-Middle income	Cultural Norms (Stigma and Discrimination), Lack of advanced HIV surveillance systems (no infrastructure to properly monitor HIV disease progression), Lack of service utilization (e.g., gaps in health care access, HIV testing, etc.)
Karamouzian et al. (2016) [33]	All Countries	The objective of this editorial was to explore and explain the barriers to collecting high-quality HIV data and generating precise HIV estimates in MENA.	Commentary	NA *	High and overlapping risk behaviors
Khajekhaze et al. (2013) [9]	Iran	The objective of this study was to assess the prevalence of HIV and related risk behaviors among PWID in Iran.	Non-Randomized Control	Lower-middle income	High and overlapping risk behaviors, Lack of service utilization (e.g., gaps in health care access, HIV testing, etc.), Cultural Norms (Stigma and Discrimination)
Khezri et al. (2022) [12]	Iran	The objective of this study was to assess changes in HIV prevalence, risk behaviors, and harm reduction utilization of PWID in Iran.	Non-Randomized Control	Lower-middle income	Cultural Norms (Stigma and Discrimination), Lack of intervention-based programs targeting PWID (Harm-reduction)
Khodayari-Zamaq et al. (2016) [34]	Iran	The objective of this study was to determine how HIV/AIDS control was initiated among policy makers’ agenda setting in Iran.	Qualitative study	Lower-middle income	High and overlapping risk behaviors, Lack of service utilization (e.g., gaps in health care access, HIV testing, etc.), Protracted Humanitarian emergencies (significant portion of a population is facing a breakdown of their livelihoods), Cultural Norms (Stigma and Discrimination), Lack of intervention-based programs targeting PWID (Harm-reduction)
Kteily-Hawa et al. (2022) [22]	All Countries	The objective of this scoping review was to establish the epidemiological HIV risk factors and underlying risk factors for youth residing in or originating from the MENA region.	Systematic Review and other Literature Reviews	NA *	High and overlapping risk behaviors, Lack of service utilization (e.g., gaps in health care access, HIV testing, etc.), Lack of advanced HIV surveillance systems (no infrastructure to properly monitor HIV disease progression), Lack of intervention-based programs targeting PWID (Harm-reduction)
Mahmud et al. (2020) [35]	All countries	The aim of the study was to delineate the epidemiology of HCV in PWID in the MENA region.	Systematic Review and Literature Reviews	NA *	High and overlapping risk behaviors, lack of service utilization (e.g., gaps in health care access, HIV testing, etc.), lack of intervention-based programs targeting PWID (Harm-reduction)
Massah et al. (2016) [36]	Iran, Iraq, Saudi Arabia	The objective of this study was to describe HIV programs in three neighboring countries (e.g., Iran, Iraq, Saudi Arabia)	Systematic Reviews and Literature Reviews	NA *	High and overlapping risk behaviors, Lack of service utilization (e.g., gaps in health care access, HIV testing, etc.), Cultural Norms (Stigma and Discrimination), Lack of advanced HIV surveillance systems (no infrastructure to properly monitor HIV disease progression), Lack of intervention-based programs targeting PWID (Harm-reduction), lack of a strong collaboration between the government, religious leaders, and public sectors
Mirzoyan et al. (2013) [37]	Libya	The objective of this study was to assess HIV prevalence and related risk factors among PWID in Tripoli, Libya.	Non-Randomized Control	Upper-Middle income	High and overlapping risk behaviors, Lack of service utilization (e.g., gaps in health care access, HIV testing, etc.), Cultural Norms (Stigma and Discrimination)
Modjarrad et al. (2017) [38]	All Countries	The objective of this commentary was to elaborate on the main points of a research article (Karamouzian) [33] detailing HIV data scarcity in MENA countries.	Commentary	NA *	High and overlapping risk behaviors, Lack of service utilization (e.g., gaps in health care access, HIV testing, etc.), Cultural Norms (Stigma and Discrimination), Lack of advanced HIV surveillance systems (no infrastructure to properly monitor HIV disease progression), Lack of intervention-based programs targeting PWID (Harm-reduction)
Moradi et al. (2016) [4]	Bahrain, Kuwait, Qatar, United Arab Emirates (UAE)	The objective of this review was to describe and synthesize information from HIV responses in Kuwait, Qatar, Bahrain and the United Arab Emirates (UAE).	Systematic Reviews and Literature Reviews	NA *	High and overlapping risk behaviors, Lack of service utilization (e.g., gaps in health care access, HIV testing, etc.), Cultural Norms (Stigma and Discrimination), Lack of advanced HIV surveillance systems (no infrastructure to properly monitor HIV disease progression), Lack of intervention-based programs targeting PWID (Harm-reduction)
Mugisa et al. (2022) [21]	All Countries	The aim of the article was to understand determinants of this region contributing to the rise in infection as well as to inform future policy implementation.	Qualitative Study	NA *	Lack of service utilization (e.g., gaps in health care access, HIV testing, etc.), Cultural Norms (Stigma and Discrimination), Lack of advanced HIV surveillance systems (no infrastructure to properly monitor HIV disease progression), Lack of intervention-based programs targeting PWID (Harm-reduction)
Mumtaz et al. (2013) [13]	Morocco	The objective of this study was to estimate the distribution of new HIV infections in Morocco by mode of exposure utilizing the MoT mathematical model.	Meta-Analysis	Lower-Middle income	High and overlapping risk behaviors
Mumtaz et al. (2014) [16]	All Countries	The objective of this study was to assess the status of the HIV epidemic among PWID in MENA by describing HIV prevalence and incidence.	Systematic Reviews and Literature Reviews	NA *	High and overlapping risk behaviors, Lack of service utilization (e.g., gaps in health care access, HIV testing, etc.), Lack of advanced HIV surveillance systems (no infrastructure to properly monitor HIV disease progression), Lack of intervention-based programs targeting PWID (Harm-reduction)
Mumtaz et al. (2014) [17]	All Countries	The objective of this study was to assess the current understanding of HIV epidemiology in the MENA region.	Commentary	NA *	Lack of advanced HIV surveillance systems (no infrastructure to properly monitor HIV disease progression)
Mumtaz et al. (2015) [15]	All Countries	The objective of this study was to assess the state of MENA region with regard to HIV and HCV.	Commentary	NA *	High and overlapping risk behaviors, Lack of service utilization (e.g., gaps in health care access, HIV testing, etc.), Cultural Norms (Stigma and Discrimination), Lack of intervention-based programs targeting PWID (Harm-reduction)
Mumtaz et al. (2018) [39]	Afghanistan, Egypt, Iran, Libya, Morocco, Tunisia	This study aimed to estimate the HIV incidence among PWID due to sharing needles/syringes in MENA.	Non-Randomized Control	NA *	Lack of intervention-based programs targeting PWID (Harm-reduction)
Mumtaz et al. (2020) [3]	All Countries	The objective of this study was to summarize information regarding knowledge and attitudes of HIV/AIDS in the MENA region.	Systematic Reviews and Literature Reviews	NA *	High and overlapping risk behaviors, Cultural Norms (Stigma and Discrimination), Lack of intervention-based programs targeting PWID (Harm-reduction), Lack of educational interventions
Mumtaz et al. (2022) [20]	All Countries	The aim of this review was to assess the trends of three key populations in the MENA region.	Systematic Reviews and Literature Reviews	NA *	Lack of service utilization (e.g., gaps in health care access, HIV testing, etc.), Cultural Norms (Stigma and Discrimination), Lack of advanced HIV surveillance systems (no infrastructure to properly monitor HIV disease progression), Lack of intervention-based programs targeting PWID (Harm-reduction)
Nasirian et al. (2012) [40]	Iran	The objective of this study was to identify the most important high-risk groups and the major MoT in Iran, as well as to compare the result with countries that ran the same model.	Systematic Reviews and Literature Reviews	Lower-Middle income	High and overlapping risk behaviors, Gaps in Healthcare Access i.e., HIV testing, Lack of advanced HIV surveillance systems (no infrastructure to properly monitor HIV disease progression), Lack of intervention-based programs targeting PWID (Harm-reduction)
Oraby et al. (2013) [41]	Egypt	The objective of the study was to explore the perspective of IDUs regarding the HIV preventive efforts.	Qualitative study	Lower-Middle income	High and overlapping risk behaviors, Lack of service utilization (e.g., gaps in health care access, HIV testing, etc.), Cultural Norms (Stigma and Discrimination), Lack of intervention-based programs targeting PWID (Harm-reduction)
Payam-Roshanfekr et al. (2021) [42]	Iran	The aim of this study was to assess the prevalence of non-injection and injection drug use and their associated factors among street-based FSWs in Iran.	Non-Randomized Control	Lower-Middle income	High and overlapping risk behaviors, Lack of intervention-based programs targeting PWID (Harm-reduction)
Rahnama et al. (2014) [43]	Iran	The objective of this study was to assess the access of PWID to harm reduction services in Tehran, Iran.	Qualitative study	Lower-Middle income	Lack of service utilization (e.g., gaps in health care access, HIV testing, etc.), Cultural Norms (Stigma and Discrimination), Lack of advanced HIV surveillance systems (no infrastructure to properly monitor HIV disease progression), Lack of intervention-based programs targeting PWID (Harm-reduction)
Seyed-Alinaghi et al. (2021) [10]	Iran	The objective of this study was to discuss the strengths and shortcomings of current HIV programs and provide suggestions to adequately respond to the ongoing HIV epidemic.	Systematic Reviews and Literature Reviews	Lower-Middle income	High and overlapping risk behaviors, Lack of service utilization (e.g., gaps in health care access, HIV testing, etc.), Cultural Norms (Stigma and Discrimination), Lack of advanced HIV surveillance systems (no infrastructure to properly monitor HIV disease progression), normalization of HIV testing
Shakiba et al. (2021) [5]	Afghanistan, Algeria, Bahrain, Egypt, Iran, Iraq, Jordan, Kuwait, Lebanon, Libya, Morocco, Palestine, Oman, Qatar, Saudi Arabia, Sudan, Syria, Tunisia, Turkey, United Arab Emirates	The objective of this study was to report epidemiological features and burdens of HIV/AIDS infection in MENA countries to develop effective policies within the region.	Systematic Reviews and Literature Reviews	NA *	Lack of service utilization (e.g., gaps in health care access, HIV testing, etc.), Cultural Norms (Stigma and Discrimination), Lack of advanced HIV surveillance systems (no infrastructure to properly monitor HIV disease progression), Lack of intervention-based programs targeting PWID (Harm-reduction)
Sharifi et al. (2018) [44]	Iran	The objective of this study was to evaluate the trend of HIV incidence among PWID, FSWs, and prisoners.	Non-Randomized Control	Lower-Middle income	High and overlapping risk behaviors
Shokoohi et al. (2016) [45]	Iran	The objective of this study was to assess HIV testing uptake and its correlates among PWID in Iran.	Non-Randomized Control	Lower-Middle income	High and overlapping risk behaviors, Lack of service utilization (e.g., gaps in health care access, HIV testing, etc.)
Springer et al. (2015) [46]	Iran	The study aimed to evaluate data available regarding women and girls globally who use or inject drugs and the impact of MAT use to treat drug addiction and prevent transmission of HIV infection among WWID.	Systematic Reviews and Literature Reviews	Lower-Middle income	Lack of service utilization (e.g., gaps in health care access, HIV testing, etc.)
Talpur et al. (2014) [18]	Iran and Iraq	The objective of this study was to review literature on the increased prevalence and consequence of heroin use and abuse in the golden crescent as well as provide suggestions toward improving drug policy in the middle east.	Systematic Reviews and Literature Reviews	Lower-Middle income	Lack of service utilization (e.g., gaps in health care access, HIV testing, etc.), Cultural Norms (Stigma and Discrimination), Lack of advanced HIV surveillance systems (no infrastructure to properly monitor HIV disease progression), Lack of intervention-based programs targeting PWID (Harm-reduction)
MENA brief report [11]	All Countries	This report aims to provide a detail report on current Harm reduction policies within the MENA region as well as the status of the HIV epidemic.	Quantitative Study	NA *	Lack of service utilization (e.g., gaps in health care access, HIV testing, etc.), Cultural Norms (Stigma and Discrimination), Lack of advanced HIV surveillance systems (no infrastructure to properly monitor HIV disease progression), Lack of intervention-based programs targeting PWID (Harm-reduction); protracted humanitarian emergencies
UNAIDS [8]	All Countries	This report aims to provide detailed statistics on the current state of the epidemic within the region pertaining to all key populations.	Quantitative Study	NA *	Lack of service utilization (e.g., gaps in health care access, HIV testing, etc.), Cultural Norms (Stigma and Discrimination), Lack of advanced HIV surveillance systems (no infrastructure to properly monitor HIV disease progression), Lack of intervention-based programs targeting PWID (Harm-reduction)

Studies are listed in alphabetical order of the author; NA *: In studies containing multiple countries, authors did not determine the income level based on the World Bank Standard; PWID: people who inject drugs; HIV: human immunodeficiency virus; UR: under-reported; MENA: Middle East and North Africa; PLHIV: people living with HIV; HCV: hepatitis C virus; IDU: injection drug users; FSW: female sex workers; MoT: modes of transmission; AIDS: acquired immunodeficiency syndrome; WWID: women who inject drugs; MAT: medication-assisted therapies.

**Table 2 ijerph-20-03844-t002:** Population size and HIV prevalence in people who inject drugs (PWID) within each country of the MENA region [8,11,35].

	People Who Inject Drugs (PWID)	
Country	Size Estimates	HIV Prevalence
Algeria	40,961 [26,333–55,590]	3.4%
Bahrain	1937 [1369–15,506]	4.6%
Jordan	4850 [3200–6500]	-
Kuwait	4050 [1850–8750]	-
Libya	4446 [2948–5943]	89.6%
Oman	4250 [2800–5700]	11.8%
Qatar	1190 [780–1600]	-
Saudi Arabia	16,800 [11,336–22,264]	9.8%
United Arab Emirates (UAE)	4800 [3200–6400]	-
Iraq	34,673 [23,115–46,230]	-
Djibouti	-	-
Egypt	90,809 [71,485–119,633]	2.6%
Lebanon	3207 [1506–4908]	0%
Morocco	18,000 [13,500–22,500]	9.6%
Somalia	-	-
Sudan	-	-
Syria	8000 [5750–10,250]	0%
Tunisia	11,000 [8462–13,750]	3.5%
Yemen	19,770 [12,710–26,830]	-
Iran	185,000 [135,000–300,000]	14%

-: No sufficient data reported.

**Table 3 ijerph-20-03844-t003:** State of epidemic in PWID within MENA countries [11,16,20].

Country	State of the Epidemic in PWID
Algeria	Concentrated
Bahrain	Concentrated
Jordan	No data
Kuwait	No data
Libya	Concentrated
Oman	Concentrated
Qatar	No data
Saudi Arabia	Concentrated
United Arab Emirates	No data
Iraq	No data
Djibouti	No data
Egypt	Concentrated
Lebanon	Low level
Morocco	Concentrated
Somalia	No data
Sudan	No data
Syria	Low level
Tunisia	Concentrated
Yemen	No data
Iran	Concentrated

## Data Availability

No new data were created or analyzed in this study.

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
