# Peer review of "An Evolving HIV Epidemic in the Middle East and North Africa (MENA) Region: A Scoping Review"

_ijerph, 2023, doi:10.3390/ijerph20053844_

Round 1

Reviewer 1 Report

The article written by Karbasi and colleagues is a scoping review summarizing the available data on HIV prevalence in people who inject drugs (PWID) in the Middle East and North Africa (MENA) region and address several issues as to why this information is scarce. The review is well written and the methods used to obtain information is clearly described and summarized. I only have some minor suggestions for the text and figures that I hope will contribute to the already high quality of this review.

Lines 57-58: “Illicit drug trafficking is a significant contributor to the HIV/AIDS epidemic.” Please indicate that this effect is indirect or adjust the sentence because the trafficking of drugs itself does not contribute to the epidemic nor does the drug itself. It is the practice of injection and the behavior around it that contribute to the epidemic, regardless of the injected substance and its legal status. This is clearly described later on in the review.

Figure 1 and accompanying text: The top left box contains a bullet point with text that is unreadable in my version of the manuscript. The box with “records excluded” states n=1229 but the text indicates that this should be 1381 incl the additional 211 duplicates (1229+211=1440). The box indicating “studies included in review” n=38 but the text states that this is 40. Please correct.

Table 1: Please indicate in table legend what each abbreviation means so that the table can be read independently of the main text (PWID, HIV, UR, MENA, NA, PLHIV, HIV, HCV, IDU). The study objective for some of the reviewed articles could perhaps be described more succinctly.

Lines 214-215 and 221-222: please use similar vocabulary to describe male and female PWID.

Lines 238-240: “ Another study observed that certain demographic characteristics contributed to increased odds of unsafe sexual and injection drug use practices, such as being married or younger.” Please indicate the age for the characteristic ‘younger’. That would also make it clear that it is not a typo because when I first read this sentence I thought it could mean ‘being married younger’.

Table 2: please indicate in the table legend what the hash indicates (-).

Figure 2: please indicate what the gradient bar indicates. Should Djubouti and Somaliland have the same color? In my version of the manuscript, Djibouti is light blue similar to countries that are not included in this review.

Lines 396-400 and table 3: This table and the text describe that the prevalence of HIV is concentrated in PWID, as compared to other demographics. However, the authors write multiple times that there is little to no data available from this region on HIV prevalence in other key populations. Therefore, I think it is too strong a statement to write that HIV is concentrated in the group of PWID because there is no solid data on other groups to support this. In my opinion, the text should be adjusted and table 3 removed.

Lines 463-465: “Iran seems to be especially proficient to combine its data with relatively high HIV testing trends among its population.” Numbers of testing in Iran, other MENA and non-MENA countries would make this interesting and relevant sentence more meaningful.

Author Response

First and foremost, thank you for your time and consideration in reviewing the article. Please see the attachment below for responses to your comments. 

Best, 
Arvin

Reviewer 2 Report

Summary: The submitted manuscript is a scoping review of knowledge gaps in the identification, intervention and care of HIV patients in PWID subpopulation in MENA countries. The authors summarize the findings of selected manuscripts and how describe the factors contributing to underreporting of data. 

Comments:

Fig 1: "Records identified from databases" cuts off at the end.

Table 1: For Esmaeli et al. (2019),Mirzoyan et al. (2013), Nasirian et al. (2012) the study objective sections seems incomplete. 

Some of the objectives in Table 1 contain abbreviations that haven't been defined.

Table 1 is an important contribution of this study. 

Is it possible to use a color coded legend to separate the papers by the problem being addressed, i.e., 1. High and Overlapping Risk Behaviors 2. Stigma and Discrimination 3. Lack of Intervention-Based Programs Targeting PWID and Service Utilization 4. Protracted Humanitarian Emergencies 5. Lack of Advanced Surveillance Systems

The authors have included other scoping reviews in their selected list of publications (Mumtaz, G.R., et al., HIV among people who inject drugs in the Middle East and North Africa: systematic review and data synthesis. PLoS Med. 2014, 11, e1001663). 

Can they add a paragraph to succinctly describe how the current manuscript improves on its predecessors?

From the Google form included in the supplementary, I can see that the authors also collected information about the study design. I propose the authors incorporate this information in the manuscript. 

For example: the number of papers selected for each study design type, or displaying this info in Table 1.

I would also recommend the authors reformat Table 1 using word-wrap and proper spacing. There is significant white space in the table, and certain lines only contain one word.

Please refer to this scoping review table as example: https://link.springer.com/article/10.1007/s10461-020-02845-x/tables/2

Author Response

First and foremost, thank you for your time and consideration in reviewing the article and providing suggestions. Please see the attachment below for the revised sections of the paper based upon your suggestions. 

Best, 
Arvin Karbasi 
